# High-Frequency Recombination of Human Adenovirus in Children with Acute Respiratory Tract Infections in Beijing, China

**DOI:** 10.3390/v16060828

**Published:** 2024-05-23

**Authors:** Fangming Wang, Ri De, Zhenzhi Han, Yanpeng Xu, Runan Zhu, Yu Sun, Dongmei Chen, Yutong Zhou, Qi Guo, Dong Qu, Ling Cao, Liying Liu, Linqing Zhao

**Affiliations:** 1Laboratory of Virology, Beijing Key Laboratory of Etiology of Viral Diseases in Children, Capital Institute of Pediatrics, Beijing 100020, China; fbi1990@126.com (F.W.); graceride@163.com (R.D.); hansir8@sina.com (Z.H.); yanpxbj@163.com (Y.X.); runanzhu@163.com (R.Z.); sunyu780312@163.com (Y.S.); dongmei_c@126.com (D.C.); 18601399785@163.com (Y.Z.); g7siete1220@163.com (Q.G.); 2Department of Critical Care Medicine, Affiliated Children’s Hospital, Capital Institute of Pediatrics, Beijing 100020, China; qudong2012@126.com; 3Department of Respiratory Medicine, Affiliated Children’s Hospital, Capital Institute of Pediatrics, Beijing 100020, China; caoling9919@163.com

**Keywords:** children, human adenovirus, co-infections, recombination, HAdV-D115

## Abstract

Recombination events in human adenovirus (HAdV) have led to some new highly pathogenic or infectious types. It is vital to monitor recombinant HAdVs, especially in children with acute respiratory tract infections (ARIs). In the retrospective study, HAdV positive specimens were collected from pediatric patients with ARIs during 2015 to 2021, then typed by sequence analysis of the penton base, hexon and fiber gene sequence. For those with inconsistent typing results, a modified method with species-specific primer sets of a fiber gene sequence was developed to distinguish co-infections of different types from recombinant HAdV infections. Then, plaque assays combined with meta-genomic next-generation sequencing (mNGS) were used to reveal the HAdV genomic characteristics. There were 466 cases positive for HAdV DNA (2.89%, 466/16,097) and 350 (75.11%, 350/466) successfully typed with the most prevalent types HAdV-B3 (56.57%, 198/350) and HAdV-B7 (32.00%, 112/350), followed by HAdV-C1 (6.00%, 21/350). Among 35 cases (7.51%, 35/466) with inconsistent typing results, nine cases were confirmed as co-infections by different types of HAdVs, and 26 cases as recombinant HAdVs in six genetic patterns primarily clustered to species C (25 cases) in pattern 1–5, or species D (1 case) in pattern 6. The novel recombinant HAdV of species D was identified with multiple recombinant events among HAdV-D53, HAdV-D64, and HAdV-D8, and officially named as HAdV-D115. High-frequency recombination of HAdVs in six genetic recombination patterns were identified among children with ARIs in Beijing. Specifically, there is a novel Adenovirus D human/CHN/S8130/2023/115[P22H8F8] designed as HAdV D115.

## 1. Introduction

Human adenoviruses (HAdVs), belonging to the family of *Adenoviridae* and the genus of Mastadenovirus, are non-enveloped, non-segmented, double-stranded DNA viruses, with a genome length of 34–36 kb and three major capsid proteins: penton base, hexon and fiber [1,2].

HAdV infections in humans are common and sometimes lethal. Until now, there were 114 types officially named and classified into seven species (A to G) based on their physicochemical, biological, and genetic characteristics. Species B, C, and E are important causative agents of acute respiratory infections (ARIs), while species D causes ocular and gastrointestinal diseases, and species A, F, and G are associated with gastroenteritis [3,4]. In 2018–2019, the incidence of HAdV infection increased sharply in children, which led to serious acute respiratory tract infections (ARIs) and possible death in many children in China, especially in the southern areas [5]. A series of guidelines for the prevention and control of HAdV infection, as well as the diagnosis and treatment of children with HAdV pneumonia, were issued by the Chinese Center for Disease Control and Prevention (http://www.chinacdc.cn/jkzt/crb/xcrxjb/201906/t20190628_203629.html, http://www.chinacdc.cn/kpyd2018/201906/t20190618_203285.html, accessed on 28 and 18 June 2019, respectively).

Among these 114 types, HAdV 52 to HAdV114 caused inconsistent serological results and misleading typing in classical HAdVs; these were typed by serum neutralization and hemagglutination inhibition tests to detect the antigenic determinants of hexon and fiber, and later typed relying on genome sequences [4]. For example, HAdV-B55 has been recognized as HAdV-B11 for decades. In an outbreak of acute respiratory disease in China, it was identified as HAdV-B11, a renal pathogen, in a serology assay, but as HAdV-B14, a respiratory pathogen, in a hemagglutination inhibition test [6]. The bioinformatics analysis of its complete genome sequence revealed that the genome consists of a HAdV-B14 backbone and partial hexon gene from HAdV-B11 [6]. HAdV-D53 isolate was originally mistyped as HAdV-D8, -D22, or -D37 on the basis of serum neutralization or limited sequencing. Genome sequence analysis revealed that the penton base, hexon, and fiber gene sequence were originated from HAdV-D37, HAdV-D22, and HAdV-D8, respectively [7]. Therefore, recombination events were common in HAdV, and were the main driver of diversity in the genus Mastadenovirus, which led to some new highly pathogenic or infectious types [4,6,8]. It is vital to monitor recombinant HAdVs, especially in children with acute respiratory tract infections (ARIs).

Moreover, penton base, hexon, and fiber genes, as well as the E3, E1 and E4 transcriptional regions, have been reported as the main recombination hotspots [9]. An important finding is that recombination, scored by characterizing the three major capsid genes, contributes substantially to the genesis of emerging pathogenic HAdVs. Therefore, the Human Adenovirus Working Group recommended that the sequences of three target genes (penton base, hexon and fiber) should be used to provide hints for novel HAdVs. Nowadays, virologists suggest that more accurate typing of HAdVs may rely on whole genome sequencing and evolutionary analysis, which can help to identify novel HAdVs [10]. In one previous study of our laboratory, the sequencing of the three capsid protein genes yielded some inconsistent results in which these capsid gene sequences were clustered into different HAdV types. However, it is hard to distinguish recombinants from co-infections [11]. 

In this study, specimens positive for HAdV were typed firstly by sequences of penton base, hexon and fiber genes. For those with inconsistent typing results for penton base, hexon and fiber genes, a modified method was developed to distinguish co-infections from the recombination of HAdV infection by using species-specific primer sets for fiber gene amplification. Then, plaque assays to purify virus isolates and meta-genomic next-generation sequencing (mNGS) for whole genome sequences were used to reveal the recombination events in children with ARIs in Beijing. 

## 2. Materials and Methods

### 2.1. Specimens Collection

Children with signs and symptoms of respiratory tract infection (such as fever, cough, chills, expectoration, nasal congestion, sore throat, chest pain, tachypnea, and abnormal pulmonary breath sounds) were diagnosed with ARIs [12]. Clinical specimens (e.g., throat swabs, sputum, and bronchoalveolar lavage fluid) were collected from children with ARIs within two days after hospitalization for an HAdV antigen test by direct immunofluorescence (DFA) test (Diagnostic Hybrids, Athens, OH, USA) or for nucleic acid detection by a capillary electrophoresis-based multiplex PCR (CEMP) assay (Ningbo HEALTH Gene Technologies Ltd., Ningbo, China). 

Pediatric patients who were positive for HAdV, younger than 14 years old, with ARIs, and admitted to the respiratory department and intensive care unit, Affiliated Children’s Hospital, Capital Institute of Pediatrics during January 2015 to December 2021, were recruited for the study. 

### 2.2. Amplification and Sequencing of Penton, Hexon, and Fiber Gene Sequence

For HAdV-positive specimens, total nucleic acids (DNA and RNA) were extracted firstly from 140 µL specimens using the QIAamp MinElute Virus Spin Kit (Qiagen GmbH, Hilden, Germany) according to the manufacturer’s instructions. Then, the penton, hexon, and fiber gene were amplified by polymerase chain reaction (PCR) using primer pairs, including penton-F (5′-CTATCAGAACGACCACAGCAACTT-3′) and penton-R (5′-TCC CGTGATCTGTGAGAGCRG-3′) to harvest the penton base gene (1153 bp), HVR-F (5′-CAGGATGCTTCGGAGTACCTGAG-3′) and HVR-R (5′-TTTCTGAAGTT CCACTCGTAGGTGTA-3′) to harvest the hexon gene (1658 bp), and fiber-F (5′-CCCTC TTCCCAACTCTGGTA-3′) and fiber-R/CR (5′-GGGGAGGCAAAATAACTACTCG-3′/ 5′-GAGGT GGCAGG-TTGAATACTAG-3′) to get the fiber gene in predicted amplicon sizes of 1153 bp, 2027 bp, 1519 bp, and 1205bp to species B, C, E and D, respectively [13], and Fiber-D1(5′-GATGTCAAATTCCTGGTCCAC-3′) and Fiber-D2 (5′-TACCCGTGCT GGTGTAAAAATC-3′) to get the fiber gene of species D in predicted amplicon size of 1205 bp [14], under the following reaction conditions: 94 °C for 5 min; 94 °C for 30 s, 52 °C for 30 s, 72 °C for 1.5 min, for 45 cycles; and 72 °C for 7 min (Table 1). PCR products were purified and sequenced using the Sanger sequencing method by Sino Geno Max Co., Ltd. (Beijing, China). The sequences were then subjected to phylogenetic analysis to identify the HAdV types. All sequences were named in the style of CHN-BJ-No./year.

### 2.3. Phylogenetic Analysis for HAdV Typing

Nucleotide sequences of penton base, hexon and fiber genes from Sino Geno Max were assembled by Lasergene’s DNA SeqMan software (version 7.1.0, DNA Star Inc. Madison, WI, USA), and subjected to BLAST analysis (https://blast.ncbi.nlm.nih.gov, accessed on 1 April 2024) to identify the most closely related genotypes of HAdVs. Multiple sequence alignment was carried out using MAFFT software. The typing of these sequences was determined by constructing phylogenetic trees with the penton base, hexon and fiber genes of reference sequences (Appendix A in attachment) retrieved from GenBank, respectively, according to the maximum likelihood method and 1000 bootstrap pseudo-replicates implemented in MEGA X with the pairwise deletion of gaps and missing data. In order to get concise phylogenetic trees, the annual representative sequences of different types were selected from sequences with high identity (>85%) and in one cluster. Homologous recombination was clearly involved in the molecular evolution of species A, B, and D, but little is known about the molecular evolution of species C. A genetic recombination analysis was carried out by Mao et al. on 201 HAdV-C strains from 9 provinces in China between 2000 and 2016 using three capsid proteins. Mao’s data revealed that phylogenetic analysis of the penton base sequences of HAdV-C revealed six genetic groups (labelled as Px1-6), which showed that the penton base had more variation than previously thought. However, multiple newly divergent sequences of species C co-circulated across mainland China [15]. Therefore, phylogenetic trees of species C were constructed more carefully by comparing them with the newly identified sequences of species C. 

### 2.4. Model Establishing to Distinguish Co-Infections from Recombination of HAdVs

Specimens with inconsistent typing results of penton base, hexon, and fiber genes were suspected of co-infections or the recombination of different types of HAdVs under the guideline of Appendix A. Based on the method of distinguishing co-infection and recombination established by McCarthy T. et al. [14], species-specific primer sets were redesigned to amplify the fiber genes of different species in separate PCR reactions (Table 1). Specimens with two different types of fiber gene sequence from the separated PCR reactions were confirmed as co-infections with different-type HAdVs, while those with only one type of fiber gene sequence were suspected of being infected by recombined HAdVs.

### 2.5. Virus Isolation

To harvest HAdV strains for plaque assays, specimens suspected of containing recombinant HAdVs were then inoculated into A549 cells, which were maintained in Dulbecco’s Modified Eagle Medium (DMEM) supplemented with 2% fetal bovine serum. The cells were observed for 5 days to record HAdV cytopathic effects (CPEs). Specimens showing typical HAdV CPE in cultures were confirmed as HAdV positive by an HAdV antigen test of DFA. And virus isolations with CPE in 75 to 80% cells were harvested and labeled as an HAdV strain. Specimens showing no CPE in three consecutive passages were negative for HAdV isolation. 

### 2.6. Plaque Assays 

Plaque assays were used to determine the titer of HAdV virus isolations that contained a possible type of recombined HAdV, then used to purify virus clones for recombination identification. 

For virus titration, 80–90% confluent A549 cells were infected with 10-fold diluted HAdV virus isolate for 2 h, then washed with PBS and covered with 3 mL of 1.2% agarose gel, which was mixed with 2 × DMEM containing FBS, 100 IU/mL penicillin and 100 μg/mL streptomycin in a ratio of 1:1. Five days later, the cells were fixed with 10% formalin and dyed with crystal violet staining solution to get the plaque forming units (pfus). Then, the multiplicity of infections (MOIs) was calculated by counting the pfu number/cell.

For virus purification, 80–90% confluent A549 cells were infected with 0.1 MOI HAdV virus isolates for 2 h, then washed with PBS and covered with 3 mL of 1.2% agarose gel, which was mixed with 2 × DMEM containing FBS, 100 IU/mL penicillin and 100 μg/mL streptomycin in a ratio of 1:1. After 5 days of incubation at 37 °C with 5 per cent CO_2_, another overlay agarose gel containing 0.01% of Finter’s neutral red was added to each well and incubated overnight. Then, plaques with agarose gel in each well were removed one by one into tubes containing 1 mL DMEM and shaken gently overnight at 4 °C. The supernatants in tubes were used to infect A549 cells and harvest enough HAdV viruses for mNGs.

### 2.7. Meta-Genomic Next-Generation Sequencing (mNGS)

Nucleic acid was extracted from purified virus clones using the QIAamp MinElute Virus Spin Kit (Qiagen GmbH, Düsseldorf, Germany) according to the manufacturer’s instructions. Then, mNGS was conducted on the Novaseq 6000 sequencing platform of Illumina (Illumina, San Diego, CA, USA), in which the 2 × 150 cycles paired-end sequencing protocol and 10 GB data yield of each library were chosen. Sequence data were trimmed using the Fastp program [16]. While human reads were removed using Bowtie2 [17], non-human reads were assembled using Trinity (version 2.8.4) and were taxonomically annotated using Kraken2 (version 2.0.8-beta) [18,19]. Finally, genomes were polished by the BLASTn package, and genome termini were manually checked and corrected using a map of the reads against the HAdV reference GenBank sequences. All sequences were named in the style of CHN-BJ-No./year.

### 2.8. Primary Recombination Analysis

Phylogenetic network analysis was performed using SplitsTree (version 4.14.6) to visualize recombination events based on the genome sequences assembled from mNGS. Potential recombination events among genome sequences were revealed using recombination detection program version 4 (RDP4) software with seven algorithms (RDP, GENECONV, Bootscan, Maxchi, Chimaera, SiSscan, and 3Seq) [20]. The reliability of the RDP4 recombination results was assessed using SimPlot software (version 3.5.1) with the parameters set to the Kimura-2-parameter model, the nucleotide conversion and transversion rate ratio to 2.0, the window to 500, and the step to 100 base pairs (bp). Default settings were used, and the threshold p-value was set at 0.05 using the Bonferroni correction for the RDP4. To avoid false-positive results, recombination events supported by at least six different methods were considered.

## 3. Results

### 3.1. HAdV Typing by sEQuencing Penton Base, Hexon and Fiber Genes

During 2015 to 2021, a total of 16,097 pediatric patients with ARIs were included; the ratio of males to females was 1.36, and the median age was 3.00 (1.00–5.95) years. Among them, 466 (2.89%, 466/16,097) were positive for HAdV determined by a DFA or CEMP assay, including 303 males and 163 females, with a median age of 2.77 (1.33–4.42) years. By the sequencing and phylogenetic analysis of the nucleotide sequences of penton base, hexon and fiber genes, there were 350 cases (75.11%, 350/466) successfully grouped into nine types, and species B, C, or E, with the most prevalent types HAdV-B3 (56.57%, 198/350) and HAdV-B7 (32.00%, 112/350), followed by HAdV-C1 (6.00%, 21/350), HAdV-E4 (1.71%, 6/350), HAdV-B14 ( 1.14%, 4/350), HAdV-C2 ( 0.86%, 3/350), HAdV-C5 (0.57%, 2/350), HAdV-C6 (0.57%, 2/350) and HAdV-B21 (0.57%, 2/350). There were 35 cases (7.51%, 35/466) with inconsistent typing results in penton base, hexon and fiber genes, which were recorded as undetermined, and 81 cases (17.38%, 81/466) lacking enough PCR products for sequencing due to a low viral load, which were recorded as unknown (Figure 1). 

B-others: HAdV-B14, -B21 et al. C-others: HAdV-C2, -C5, C6 et al. Undetermined: specimens with different typing results of the three capsid genes. Unknown: specimens lacking at least one gene sequence among the three capsid genes.

### 3.2. Co-Infection of HAdVs Identified on the Basis of Fiber Gene Sequences

By the model distinguishing co-infections from the recombination of HAdVs on the basis of fiber gene sequences, there were nine cases co-infected by different types of HAdVs, with 66.67% (6/9) co-infected by types of species B and C, and 33.33% (3/9) co-infected by type 3 and 7 of species B (Table 2). The other 26 cases were confirmed with single-type fiber gene sequence, which were then analyzed deeply for recombinant identification.

### 3.3. Primarily Recombination Analysis on the Basis of Penton Base, Hexon and Fiber Gene Sequences

With the nine co-infection cases excluded from those 35 cases defined as undetermined with inconsistent typing results in penton base, hexon and fiber genes, there were 26 cases where recombination HAdVs were suspected, including 25 cases primarily clustered to species C, and 1 to species D with partial of them shown in Appendix A.

These phylogenetic ML (maximum likelihood) trees of species C were constructed on the basis of nucleotide sequences of penton base, hexon and fiber gene sequences (Figure 2). The fiber gene sequences of the 25 cases were clustered into two clades, F2 and F5, while the hexon gene sequences were clustered into three clades H1, H2, and H5, and the penton base gene sequences exhibited high variability in three major clades (Px1ps1, Px3 and P89) with a mean genetic distance of 1.2%.

The hexon gene sequence of the case belonging to species D (CHN-BJ-S8130/2021) shared the highest identity with that of HAdV-8 and -54, while its penton base gene sequence shared high identity with that of HAdV-64, -42, and -22, and its fiber gene sequence was more similar to that of HAdV-85, -8 and-53 (Figure 3).

Recombining all the typing results of the penton base, hexon and fiber genes of those 26 cases, there were six genetic patterns from 1 to 6 shown in Table 1, including five belonging (genetic pattern 1–5) to species C and one (genetic pattern 6) belonging to species D (Table 3). 

### 3.4. Recombination Analysis on the Basis of the Whole Genome Sequences

The specimens suspected of being infected with recombinant HAdVs were cultured in A549 cells, and five virus isolates belonging to genetic pattern 2–6, respectively, were successfully harvested and purified by plaque assays. The whole genome sequences of CHN-BJ-86413/2017, CHN-BJ-93578/2018, CHN-BJ-95031/2018, CHN-BJ-1w5060/2019, and CHN-BJ-S8130/2021 from mNGS were assembled in non-parametric style, and combined with parametric assembly optimization by using MZ151862_RUS-Novosibirsk_8.234V_2020, MK041234_Shanxi-CHN_22_2002, MK041237_Shanxi-CHN_38_2007, MK165453_CHN-SX-2004-327_2004 and FJ169625_DEU_IAI-1_2005 as reference sequences, respectively, for phylogenetic analysis. The raw data for meta-genomic next-generation sequencing were submitted to GenBank with the numbers of SRR28980648 (CHN-BJ-86413/2017), SRR28980649 (CHN-BJ-93578/2018), SRR28980647 (CHN-BJ-95031/2018), SRR28980650 (CHN-BJ-1w5060/2019) and SRR28980646 (CHN-BJ-s8130/2021), respectively. Then, their genome sequences also were submitted to GenBank with the numbers of PP786308(CHN-BJ-86413/2017), PP786307 (CHN-BJ-93578/2018), PP786310 (CHN-BJ-95031/2018), PP786309 (CHN-BJ-1w5060/2019) and OR044915 (CHN-BJ-s8130/2021), respectively. All these sequences will be released after the publication of the manuscript.

In the phylogenetic tree of complete HAdV-C genomes containing reference sequences from GenBank, four strains isolated in this study, representing genetic patterns 2-5, respectively, were grouped into two clusters; cluster C2 containing clade 2 strain CHN-BJ86413/2017 shared high similarity with the Egypt strain isolated in 2002, and clade 3 strain CHN-BJ-1w5060/2019 closely related to the Dandong strain in China, and cluster C5 containing clade 2 strains CHN-BJ-93578/2018 and CHN-BJ-95031/2018 closely related to strains isolated from Shanxi in different periods (Figure 4).

To explore the breakpoints of these four HAdV-C isolates, the Simplot software tool was used. The results revealed that multiple recombination events existed among the four HAdV-C isolates from the whole genome sequences of HAdV-1, HAdV-2 and HAdV-5 (Figure 5) with strain CHN-BJ-86413/2017 typed as C89, CHN-BJ-1w5060/2019 as C104, and CHN-BJ-93578/2018 and CHN-BJ-95031/2018 unidentified (Appendix A).

The complete HAdV-D genome of the strain (CHN-BJ-S8130/2021 isolated from a 21-day-old neonate with a complete genomic sequence of 35,131 bp and a GC content of 56.15% showed a closer relationship with HAdV-D53 and D85 in phylogenetic analyses containing reference sequences from GenBank (Figure 6). SimPlot analysis revealed that CHN-BJ-S8130/2021 was determined as HAdV-P22H8F8 concerning the sequence of the penton base, hexon, and fiber genes, a novel recombinant genotype according to the current nomenclature rules based on the three major capsid genes. 

Analyzed by the Simplot software tool, two recombinant events were identified in the genome sequence of CHN-BJ-S8130/2021 involved with HAdV-D53, HAdV-D64 and HAdV-D8, which were confirmed by seven methods implemented in the RDP package (KAP-Value 1.22 × 10^−160^, 2.83 × 10^−217^) (Appendix A). Therefore, strain CHN-BJ-S8130/2021 was a novel recombinant HAdV with the major region of 11,444–17,721 bp containing the penton base gene originated from HAdV-D64 (P22H19F37), while the minor region of 17,722–20,998 bp containing the hexon gene was derived from HAdV-D8, and the backbone was from HAdV-D53 (P37H22F8) with the fiber gene from AdV-D8 (Figure 7).

This novel HAdV-D genome of CHN-BJ-S1830/2021, identified from the respiratory specimens of a 24-day-old neonate, was submitted to GenBank and given an accession number of OR044915. The Human Adenovirus Working Group checked the genome sequence and designated the novel Adenovirus D human/CHN/S8130/2023/115[P22H8F8] as HAdV D115.

## 4. Discussion

Recombination is a common way of virus evolution, following base substitutions and insertion/deletion. Homologous recombination of the penton base, hexon and fiber genes appears to be a major mechanism of HAdV evolution. Therefore, classical HAdV typing methods detecting the antigenic determinants of hexon and fiber by serum neutralization or hemagglutination inhibition have a rather poor chance to distinguish co-infections from recombinant HAdV infection; the PCR method, only amplifying the partial hexon gene, is also not appropriate for HAdV recombinant identification. Wu et al. used universal PCR and sequencing primers for the penton base, hexon, and fiber gene in HAdV typing [13]. In the retrospective study, specimens positive for HAdV from pediatric patients with ARIs during 2015 to 2021 were typed on the basis of penton base, hexon and fiber gene sequences. The results revealed that HAdV-B3 (56.57%, 198/350) and HAdV-B7 (32.00%, 112/350), followed by HAdV-C1 (6.00%, 21/350), were common HAdV types associated with respiratory tract infections in children in Beijing, China. However, there were 35 specimens with inconsistent typing results in penton base, hexon and fiber genes (7.51%, 35/466), which were suspected of being co-infected by multiple types of HAdV or infected by recombinant HAdV. 

McCarthy et al. proposed a clinical algorithm for detecting the co-infections of HAdVs by PCR amplification and the sequencing of the sub-regions of the hexon and fiber genes [14]. However, phylogenetic analysis of the penton base gene sequences of HAdV-C indicated that the penton base has more variations. And recombination events in the penton base gene of strains belonging to species C are likely to be missed. In this study, we revised the method to distinguish co-infections from the recombination of HAdVs using two species-specific primer sets for fiber gene amplification. Accordingly, nine of these 35 cases were confirmed as a co-infection of HAdVs and other 26 cases were suspected of being infected by recombinant HAdVs with genetic pattern 1–6, including five belonging (genetic pattern 1–5) to species C and one (genetic pattern 6) belonging to species D. 

Although co-infections of HAdVs with other viruses are very common, data about the co-infections of different HAdV types were accumulated gradually following the development of PCR-based identification targeting hexon or fiber genes in the identification of co-infections [14,21,22]. In Taiwan, 25 (14.3%) of 175 children with ARIs were co-infected by different HAdV types, with co-infections of HAdV-3 and HAdV-2 being common [21]. However, a report about the repeated infections of HAdVs in immunocompromised patients and the multiple strains of HAdVs causing concomitant infection in US military trainees illustrated the complexities of evaluating HAdV infections [23]. In this study, the co-infections of species B (HAdV-B7) and C (HAdV-C2) are more common (66.67%, 6/9) than those of HAdV-B3 and B7 in species B (33.33%, 3/9). It is more likely that synergistic, competitive, and inhibitory effects between pathogens resulted in specific infection patterns [24]. HAdV persistence infections allow for the continuous production of a progeny virus over an extended period, which increases the chance of co-infections [25].

In order to confirm the recombinant events of HAdVs, five strains belonging to genetic patterns 2–6 of recombination were successfully isolated, then purified by plaque assays to get cloned and sequenced by mNGS for the whole genome sequences. There were four strains belonging to HAdV-C and one to HAdV-D, determined by RDP4 analyses.

HAdV-C infection usually causes mild diseases [26]. However, it is associated with severe diseases and even epidemic outbreaks among organ transplant populations, or with persistently staying in the lymphatic system, such as the tonsils or intestinal lymph nodes, after infection [3,27]. HAdV-C infection has the potential to be reactivated when the body’s immunity declines [28]. As shown by recent data, multiple recombination events in all HAdVs have given rise to several new recombinant isolates. Some of them may be associated with severe respiratory illness [15,29,30,31,32]. By the recombination analysis of published HAdV-C genome sequences, Rivailler et al. [32] have identified 20 sequences of HAdV-C and two main regions where HAdV-C recombination occurs, specifically within the hexon gene and near the fiber gene region. Mao et al. [15] reported at least 16 genetic patterns of HAdV-C recombination among 201 HAdV-C strains from 9 provinces in China between 2000 and 2016 on the basis of the gene sequences of three capsid proteins. However, the recombinant isolates reported were not recognized as new genotypes, for the recombination events brought no significant change in their biological characteristics. Although researchers suggested that the penton base of HAdV-C with nucleotide diversity is 0.008, Mao et al. revealed that the penton base of HAdV-C may be more genetically diverse than previously believed, despite its genetic distance of 1.2%. So far, the Human Adenovirus Working Group has officially named four novel types of HAdV-C, including HAdV-C57 (P1H57F6), HAdV-C89 (P89H2F2), HAdV-C104 (P1H1F2) and HAdV-C108 (P1H2F2) (http://hadvwg.gmu.edu/, accessed on 1 April 2024). In the research work, multiple recombination events were identified among the four HAdV-C isolates, with one typed as C89, one as C108, and two unidentified. More evidence should be gathered to assess the clinical effects of these recombination events.

It is noteworthy that a novel HAdV-D genome of CHN-BJ-S1830/2021 was identified from the respiratory specimens of a 24-day-old neonate. The phylogenetic tree of whole genome sequences showed that CHN-BJ-S1830/2021 was clustered together with HAdV-D53 and HAdV-D85. By the phylogenetic tree construction and recombination analysis of capsid protein gene sequences, it is supposed that the recombinant events might have occurred among HAdV-D53, HAdV-D64, and HAdV-D8. HAdV-D usually has an ocular or gastrointestinal tropism while its respiratory association is scarcely reported. Furthermore, HAdV-D53, HAdV-D64, and HAdV-D8 were major EKC pathogens. Some data have shown a correlation between fiber knob amino acids and corneal tropism on the basis of the genomic analysis of a large set of currently and historically important human adenovirus pathogens [33]. Many HAdV strains causing EKC can be isolated from the urinary system, and vertical transmission from mother to children cannot be ruled out. It is similar to HAdV-D56, a recombinant one from types causing keratoconjunctivitis (EKC), isolated in France and determined as the pathogen of a dead newborn with severe respiratory infections, which were transmitted to three healthcare workers [34]. It can be presumed that the co-infections of HAdVs in the genitourinary tract promote the intra-species recombination of HAdV-D to form a novel type. Therefore, the isolated CHN-BJ-S8130/2021 is a recombinant virus from several previously identified types of HAdVs, with its respiratory tropism different from its parent virus strains HAdV-D 64 and -8, which cause corneal conjunctivitis; furthermore, the virus conforms to the standard of a novel type of HAdV-D. The isolated CHN-BJ-S8130/2021 was officially named D115 by the Human Adenovirus Working Group on 26 February 2024.

## 5. Conclusions

In summary, a modified method to distinguish co-infections from recombination was developed in the research work, and six genetic recombination patterns of HAdVs were identified in children with ARIs in Beijing between 2015 and 2021; five of them were confirmed by whole genome sequences from mNGS. These data suggested the high frequency of the recombination of HAdVs among children with ARIs in Beijing. Moreover, a novel recombinant type HAdV-D115, found in this study, was officially named.

## Figures and Tables

**Figure 1 viruses-16-00828-f001:**
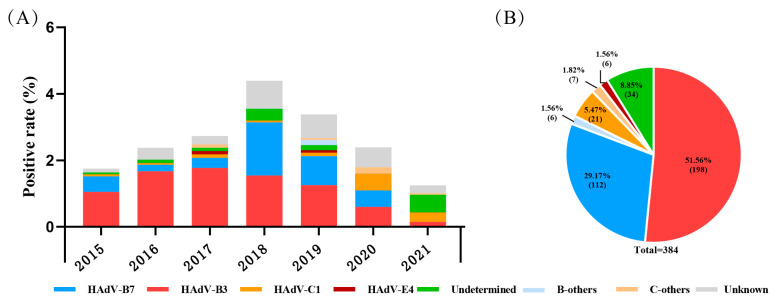
The distribution of HAdV-positive cases in pediatric patients with acute respiratory illness from January 2015 to December 2021 in Beijing. (**A**) The yearly distribution of HAdV-typing results. (**B**) Proportions of different HAdV types.

**Figure 2 viruses-16-00828-f002:**
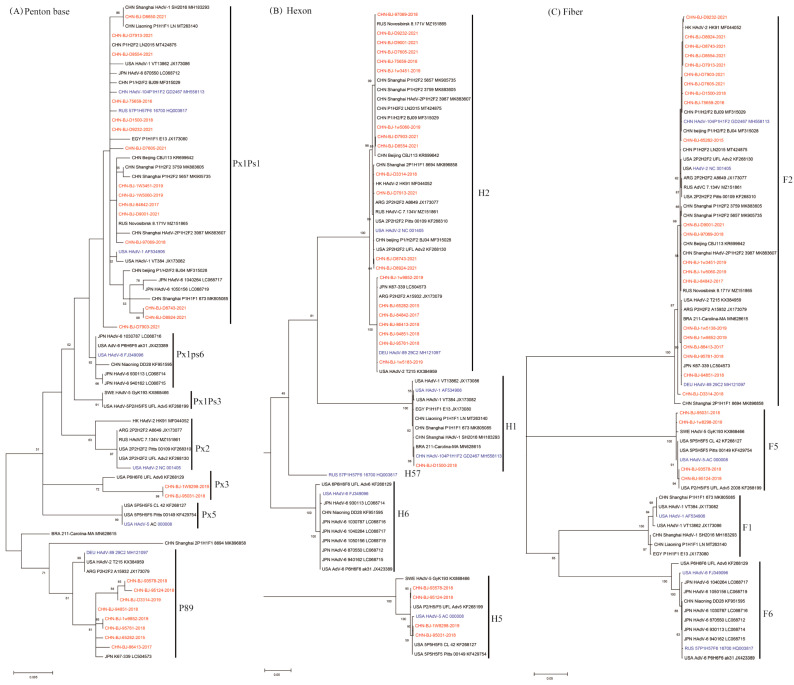
The ML trees constructed on the basis of the nucleotide sequences of penton base, hexon, fiber genes from the 25 cases compared to those of species C from GenBank using the maximum likelihood method of MEGA X with 1000 bootstraps. Sequences from the study were shown in red color, while prototype sequences belonging to novel types from GenbBank were in blue, and others from GenBank in black. (**A**) The phylogenetic tree of the nucleotide sequences of the penton base gene, divided into clusters of Px1Ps1, Px1Ps6, Px1Ps3, Px2, Px3, Px5 and P89. Ps, the subgroup. Px, the major genetic group. (**B**) The phylogenetic tree of the nucleotide sequences of hexon gene, divided into H1, H2, H57, H6 and H5. (**C**) The phylogenetic tree of the nucleotide sequences of the fiber gene, divided into F2, F5, F1 and F6.

**Figure 3 viruses-16-00828-f003:**
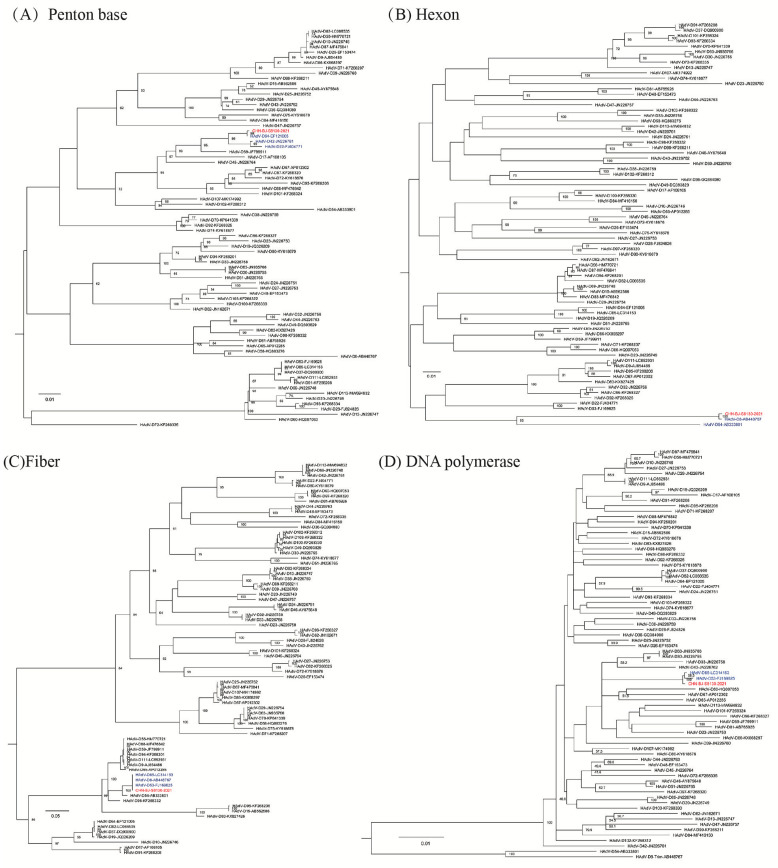
The ML (maximum likelihood) trees constructed on the basis of the nucleotide sequences of the penton base (**A**), hexon (**B**), fiber (**C**) and DNA polymerase (**D**) gene sequences of the isolation CHN-BJ-S8130/2021 belonging to species D and those from GenBank using the maximum likelihood method of MEGA X with 1000 bootstraps. The sequence from the study was shown in red color.

**Figure 4 viruses-16-00828-f004:**
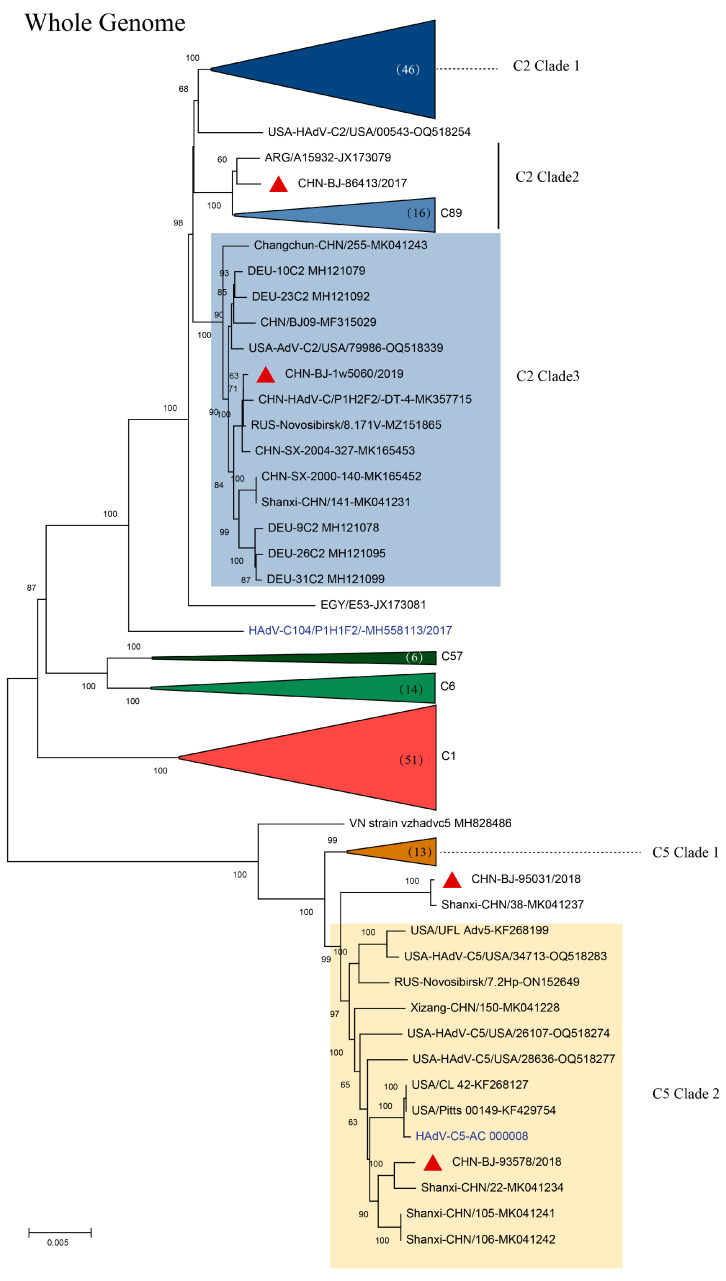
The ML (maximum likelihood) tree of the whole genome sequences of the four strains isolated in this study and 180 whole genome sequences of HAdV-C from GenBank constructed using the maximum likelihood method of MEGA X with 1000 bootstraps. Red triangles in clusters C2 clade 3 and C5 clade 2 represent sequences from the study.

**Figure 5 viruses-16-00828-f005:**
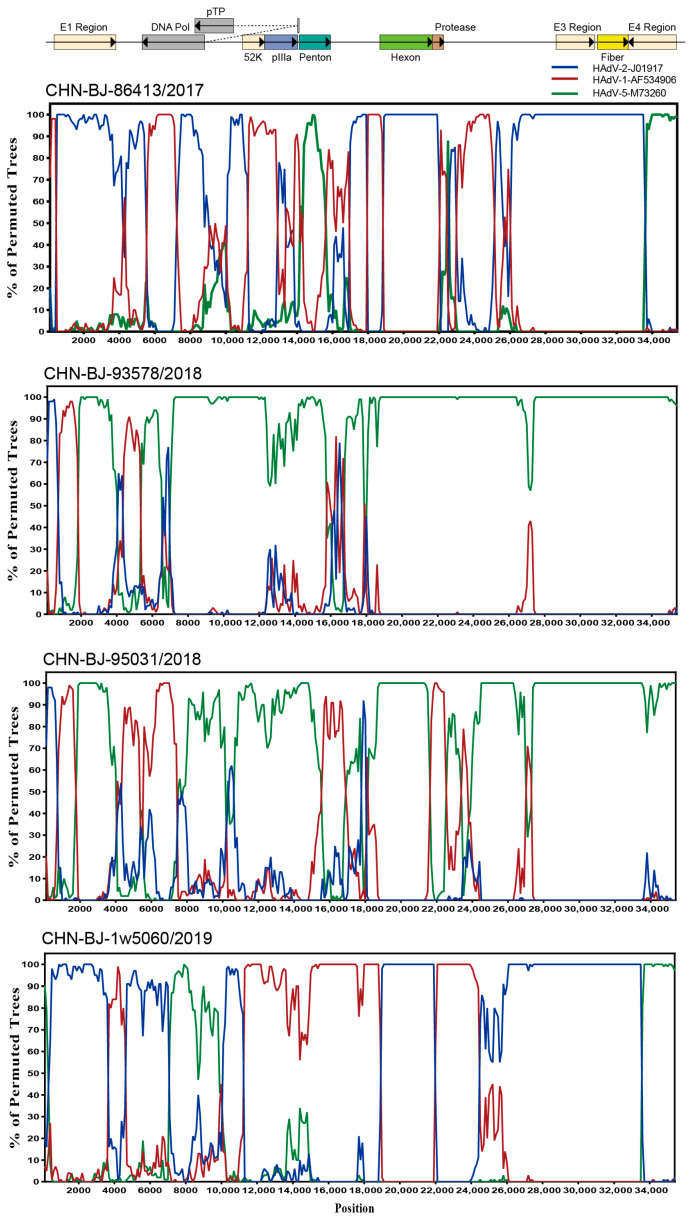
Bootscanning analysis of the whole genome sequences of the four HAdV-C isolates by Simplot software tool. Graphs were generated using CHN-BJ-86413/2017, CHN-BJ-93578/2018, CHN-BJ-95031/2018 and CHN-BJ-1w5060/2019 as query sequences. HAdV-2-J01917 (prototype) was shown in blue color. HAdV-1-AF534906 (prototype) was shown in red color. HAdV-5-M73260 (prototype) was shown in green color. The bootstrp value is 500 bp for a window with a 100 bp step.

**Figure 6 viruses-16-00828-f006:**
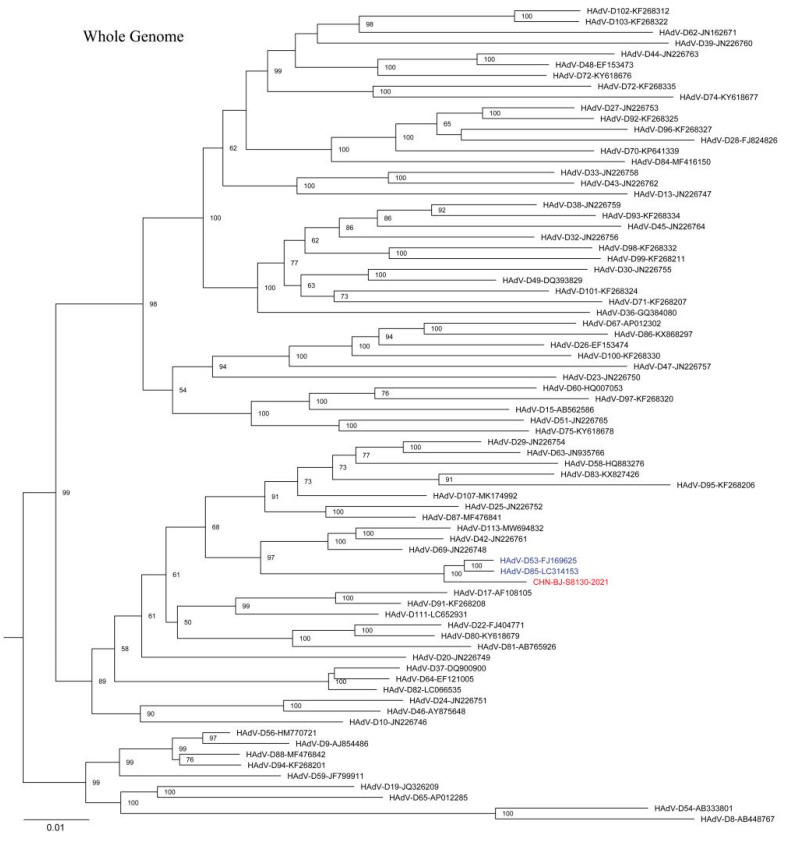
The ML tree constructed on the basis of the whole genome sequence of the isolation CHN-BJ-S8130/2021 and sequences from GenBank using the maximum likelihood method of MEGA X with 1000 bootstraps. The sequence from the study was shown in red color.

**Figure 7 viruses-16-00828-f007:**
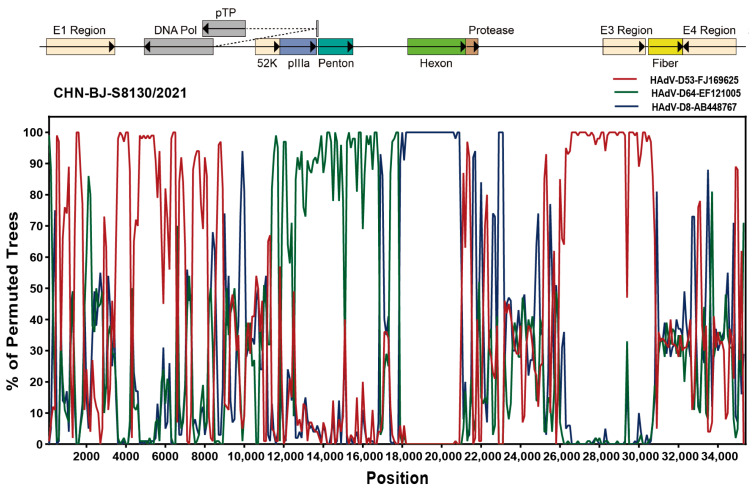
Bootscanning analysis of the whole genome sequences of the strain CHN-BJ-S8130/2021 by the Simplot software tool.

**Table 1 viruses-16-00828-t001:** Primers designed for penton base, hexon and fiber gene amplification, especially those for the fiber gene of different species.

Primer Pairs	Forward/Reverse Primer (5′–3′)	Length of Amplicons
Penton F	CTATCAGAACGACCACAGCAACTT	1253 bp
Penton-R	TCCCGTGATCTGTGAGAGCRG	
HVR-F	CAGGATGCTTCGGAGTACCTGAG	1685 bp
HVR-R	TTTCTGAAGTTCCACTCGTAGGTGTA	
Fiber-F	CCCTCTTCCCAACTCTGGTA	
Fiber-R	GGGGAGGCAAAATAACTACTCG	B (1153 bp)/E (1519 bp)
Fiber-CR	GAGGTGGCAGGTTGAATACTAG	C (2027 bp)
Fiber-D1	GATGTCAAATTCCTGGTCCAC	
Fiber-D2	TACCCGTGCTGGTGTAAAAATC	D (1205 bp)

**Table 2 viruses-16-00828-t002:** Specimens co-infected by different types of HAdVs on the basis of fiber gene sequences.

No. of Specimens	Type of Penton Base (P)	Type of Hexon (H)	Type of Fiber (F)	Collection Year
BJ 73148	P1	H3	F2/3	2016
BJ 92087	P7	H3	F7/3	2018
BJ 95131	P1	H7	F7/2	2018
BJ 95634	PC (double peaks) *	H7	F7/2	2018
BJ 96125	P1	H7	F7/2	2018
BJ 1w799	P1	H7	F7/2	2018
BJ D1458	P3	H7	F7/3	2018
BJ D1465	P7	H3	F7/3	2018
BJ 1w6838	PC *	H2	F7/2	2019

* The type of penton base gene is HAdV-C.

**Table 3 viruses-16-00828-t003:** Genetic patterns identified on the basis of penton base, hexon and fiber gene sequences in children with ARIs in Beijing.

Genetic Patterns	No. of Cases	Type of Penton Base *	Type of Hexon	Type of Fiber	Collection Year	Known Type
1	1	Px1Ps1	H1	F2	2018	C104
2	7	P89	H2	F2	2017~2019	C89
3	14	Px1Ps1	H2	F2	2015~2021	Maybe C108
4	1	P89	H5	F5	2018	NA
5	2	Px3	H5	F5	2018, 2019, 2021	NA
6	1	P64 or 22	H8	F8	2021	NA

* The nucleotide sequences of the penton base gene, divided into clusters of Px1Ps1, and Px3. Ps, the subgroup. Px, the major genetic group. Ps, the subgroup. Px, the major genetic group.

## Data Availability

The datasets used and/or analysed during the current study are available from the corresponding author on reasonable request. All the sequences obtained during this study were submitted in GenBank under accession numbers PP786307-PP786310 and OR044915.The raw data for Meta-genomic Next-Generation Sequencing in this study have been deposited in the GenBank under accession numbers SRR28980646- SRR28980650.

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
