# Peer review of "High-Frequency Recombination of Human Adenovirus in Children with Acute Respiratory Tract Infections in Beijing, China"

_viruses, 2024, doi:10.3390/v16060828_

Round 1
Reviewer 1 Report
Comments and Suggestions for Authors
High-frequency recombination of HAdV in hospitalized children with acute respiratory tract infections in Beijing, China. Wang et al. have aimed to investigate the recombination frequency of HAdVs in hospitalized children with acute respiratory tract infections (ARIs) during 2015-2021 in Beijing. To this end, HAdVs were typed by sequencing the genes for structural proteins hexon, penton base and fiber. If inconsistent typing results were obtained a modified method with species -specific primer sets for fiber was developed to distinguish co-infections from infections by recombinant HAdVs that was followed by mNGS. From the 466 HAdV+ cases the great majority was typed as B3 (56.6%) and B7 (32%), 6% as HAdV-C1 and a few other C-types. Of the 35 cases with inconsistent typing results, 9 were considered as co-infections, 26 as potential recombinant HAdVs differentiated in 6 patterns, mostly among HAdV-C. Interestingly, one type was identified as a novel recombinant of species D with genomic parts derived from HAdV-D53 (the backbone), D64 (11,444-17,721bp) and from D8 (17,722-20998 bp) resulting in the following pattern of capsid proteins (P64H8F8), now officially named as HAdV-D115.
The topic was introduced reasonably well, and the methods were described in sufficiently good detail. The results are well described with a few exceptions:
Fig. 4: The color code of the boxes on top of the bootscanning analysis should be explained. The discussion is largely comprehensive, yet is in my view at times a little oversimplifying, e.g. linking the disease association of EKC solely with different receptor interactions. It is also possible that several HAdV types infect the same tissue, but only a few may cause disease, possibly due to different immunomodulatory functions in the E3 unit.
The quality of English is mostly very good. Occasionally, there are some minor errors though that need to be corrected which the reviewer has mostly done already (see below).
L54: Add here a few refs for novel HAdVs proven to be caused by recombination e.g. ref 6 (B55=B14+B11 hexon) and ref. 7 (D53 origin).
L78: Instead of writing, In the study, use: In this study, when you refer to your own work (see also L292, L305, L318
L135: fiber gene sequence
L165/6: sentence is not correct!! …were purified and virus clones used for infection of A549 cells….?
L209: Primary recombination analysis…
L217: …high variability…
L220: ML should be properly introduced at first use, (maximum likelihood) etc.
L230/1: as above, should be this. The sequence from this study
Table 1 should be on one page.
L239: Start with capital letter; Specimens..
L257:..four HAdV-C isolates from…
L279/280: …while the backbone was from…
L289: …inhibition have a rather poor chance to distinguish co-infections…
L299: I am not sure whether 7.5% is a high frequency of co-infection or recombinant..
L307: Accordingly, nine of these 35…
L319: ..than those of HAdV-B3..
L332: I think this applies essentially to all HAdVs.
L350: ..should be gathered..
L356: …might have occurred …

The quality of English is mostly good. Occasionally, there are some minor errors in the English that need to be corrected. These are indicated in the reviewer-corrected ms attached.
Author Response
The topic was introduced reasonably well, and the methods were described in sufficiently good detail. The results are well described with a few exceptions:
Fig. 4: The color code of the boxes on top of the bootscanning analysis should be explained. The discussion is largely comprehensive, yet is in my view at times a little oversimplifying, e.g. linking the disease association of EKC solely with different receptor interactions. It is also possible that several HAdV types infect the same tissue, but only a few may cause disease, possibly due to different immunomodulatory functions in the E3 unit.
Response: Thanks for your advice. To make the title more accurate, we revised the explanation of Fig. 5“Bootscanning analysis of the whole genome sequences of the four HAdV-C isolates by Simplot software tool. Graphs were generated using CHN-BJ-86413/2017, CHN-BJ-93578/2018, CHN-BJ-95031/2018 and CHN-BJ- 1w5060/2019, as query sequences. HAdV-2-J01917 (prototype) was shown in blue color. HAdV-1-AF534906 (prototype) was shown in red color. HAdV-5- M73260 (prototype) was shown in green color. The bootstrp value is 500bp for a window with 100bp step.”
And the discussion concerning on “The CHN-BJ-S8130/2021 strain shared high similarity to HAdV-D8 on the fiber gene sequences which can bind to common corneal epithelial receptors (CD46 or sialic acid), and determine the tropism for cornea [32]” was revised to“HAdV-D usually has an ocular or gastrointestinal tropism while its respiratory associa-tion is scarcely reported. Furthermore, HAdV-D53, HAdV-D64, and HAdV-D8 were major EKC pathogens. Some data have shown a correlation between fiber knob amino acids and corneal tropism on the basis of genomic analysis of a large set of currently and historically of important human adenovirus pathogens”.
The quality of English is mostly very good. Occasionally, there are some minor errors though that need to be corrected which the reviewer has mostly done already (see below).
L54: Add here a few refs for novel HAdVs proven to be caused by recombination e.g. ref 6 (B55=B14+B11 hexon) and ref. 7 (D53 origin).
Response: As you suggested, we changed several references.
L78: Instead of writing, In the study, use: In this study, when you refer to your own work (see also L292, L305, L318
Response: We changed all “In the study” to “In this study”.
L135: fiber gene sequence
Response: We edited it.
L165/6: sentence is not correct!! …were purified and virus clones used for infection of A549 cells….?
Response: The sentence was revised to “The supernatants in tubes were used to infect A549 cells and harvest enough HAdV viruses for mNGs.”
L209: Primary recombination analysis…
L217: …high variability…
L220: ML should be properly introduced at first use, (maximum likelihood) etc.
L230/1: as above, should be this. The sequence from this study
Table 1 should be on one page.
L239: Start with capital letter; Specimens.
L257:..four HAdV-C isolates from…
L279/280: …while the backbone was from…
L289: …inhibition have a rather poor chance to distinguish co-infections…
L299: I am not sure whether 7.5% is a high frequency of co-infection or recombinant..
L307: Accordingly, nine of these 35…
L319: ..than those of HAdV-B3..
L332: I think this applies essentially to all HAdVs.
L350: ..should be gathered..
L356: …might have occurred …
Response: All these errors were corrected.
Comments on the Quality of English Language
The quality of English is mostly good. Occasionally, there are some minor errors in the English that need to be corrected. These are indicated in the reviewer-corrected ms attached.

Reviewer 2 Report
Comments and Suggestions for Authors
In this paper, Wang and De et al. describe the isolation of human adenovirus types, including co-infections and recombinants, from clinical samples from children hospitalized for acute respiratory tract infections in Beijing, China, during 2015-2021. A majority of the isolates were Species B (Ad3 and Ad7); HAdV-C1 was the third most common isolate. The article emphasizes the need to go beyond routine traditional neutralization and hemagglutination assays. Extensive work was done to determine the adenovirus type for Penton, Hexon, and Fiber for these virus isolates in addition to plaque-purifying viruses with discrepancies to identify co-infections and recombinants. Sequences were analyzed to determine phylogenetic trees and bootscanning to determine cross-over points in genome sequences. It is reasonable to determine new recombinants, especially if there may be changes in phenotypic properties in the recombinant, as was seen in the Species D recombinant isolated (which has not been catalogued as a new adenovirus type (HAdV-D115).
I think the title of the article may be a bit misleading. It makes it seem that the recombination events are happening within the hospital setting. Perhaps it is more accurate to say “High-frequency isolation of human adenovirus recombinants in hospitalized children with acute respiratory tract infections in Beijing, China”.
The presentation of Tables in the article is questionable. The first Table may have originally been a Supplemental Table (it is labeled as S2), and then is followed in order by Table 2 and Table 1. Tables really should occur sequentially in the text. And the Supplemental Table S1 had misaligned portions when I downloaded and printed it (overlapping text on several pages).
I am somewhat concerned about the plaque assay methods. My own experience would suggest that fixing a plaque assay at day 5 will give a much lower titer than would occur a few days later (especially for HAdV-B3 and HAdV-B7). This would affect your MOI calculations so that the 0.1 MOI would be an underestimate of the true titer. It would be good to describe your plaque purification better. You describe it as an MOI of 0.1, but that would not truly be reasonable in picking plaques after 5 days when a plaque would involve dozens of cells (unless you mean a MOI that would give less than 1 plaque per small well). It just is not a clear enough description.
If it is speculated that co-infections or recombinants lead to more severe disease, was there any clinical evidence for that occurrence (or references that would indicate that)? Was there any evidence that any of the recombinants were circulating in the community?
Some technical issues with Figures and Tables in the article:
The text in the Figures was too small to be legible (Figures 1, 2, and 5). Even with magnification, they became pixelated and were difficult to read. Perhaps in the published article the print will be more readable, but that may require a better initial figure before the decrease in text size.
Table 1 needs to be on a single page (in the current manuscript the column headings are on a separate page). It would be helpful to have a better or more extensive description of the categories that are used for Penton. While there is a reference for this, it was in a less accessible journal, so I was not able to see what the differences are.
In line 255, “breakthrough points” may not be the best term. Are these “break points” or “recombination points” instead? Similarly, in line 279, the term “bone frame” is not clear to me. It this the “backbone” or “framework” or some other description?
Although 4 or 5 isolates were Species C recombinants, there is not a suggestion that these had identifiable “changes in biological characteristics”. The more interesting observation was that the Species D recombinant was substantially different from the parental types, and the classification as a new Species D type (HAdV-D115) is a very important part of the paper. It may be getting less attention in the paper than it deserves.
Comments on the Quality of English LanguageI have noted a few changes in the text above.
You may want to polish the English usage (with someone whose first language is English). While the text is quite readable, there are some subtle ways in which the language could be improved.
Author Response
- I think the title of the article may be a bit misleading. It makes it seem that the recombination events are happening within the hospital setting. Perhaps it is more accurate to say “High-frequency isolation of human adenovirus recombinants in hospitalized children with acute respiratory tract infections in Beijing, China”.
Response: Thanks for your advice. To make the title more accurate, we revised it to “High-frequency recombination of human adenovirus in children with acute respiratory tract infections in Beijing, China”.
- The presentation of Tables in the article is questionable. The first Table may have originally been a Supplemental Table (it is labeled as S2), and then is followed in order by Table 2 and Table 1. Tables really should occur sequentially in the text. And the Supplemental Table S1 had misaligned portions when I downloaded and printed it (overlapping text on several pages).
Response: Sorry for the wrong presentation of Tables.
Table S1 is in “2.3. Phylogenetic analysis for HAdV typing”, with the heading of “Table S1 The reference genome sequences downloaded from GenBank”. For it is too large to be included in manuscript, we will submit it in attachment.
Table S2 in “2.4. Model establishing to distinguish co-infections from recombination of HAdVs” with the heading of “Table S2. Species-specific primer sets designed according to the fiber gene sequences of different species.” was edited to “Table 1. Primers designed for hexon, penton base and fiber gene amplification, especially those for fiber gene of different species.”
The Table 2 was in “3.2. Co-infection of HAdVs identified on the basis of fiber gene sequences” with the heading of “Table 2. Specimens co-infected by different types of HAdVs on the basis of fiber gene sequences.”
The Table 1 in “3.3. Primarily recombination analysis on the basis of hexon, penton base and fiber gene sequences” should be Table 3, with the heading of “Table 3. Genetic patterns identified on the basis of penton base, hexon and fiber gene sequences in children with ARIs in Beijing.”
- I am somewhat concerned about the plaque assay methods. My own experience would suggest that fixing a plaque assay at day 5 will give a much lower titer than would occur a few days later (especially for HAdV-B3 and HAdV-B7). This would affect your MOI calculations so that the 0.1 MOI would be an underestimate of the true titer. It would be good to describe your plaque purification better. You describe it as an MOI of 0.1, but that would not truly be reasonable in picking plaques after 5 days when a plaque would involve dozens of cells (unless you mean a MOI that would give less than 1 plaque per small well). It just is not a clear enough description.
Response: Thanks for sharing your experience. It’s true that fixing a plaque assay at day 5 may give a much lower titer than would occur a few days later. In the manuscript, we only picked some plaques (≥3) for amplifying and sequencing. In future experiments, we will pay more attention to the questions you raised and get more accurate titer.
4.If it is speculated that co-infections or recombinants lead to more severe disease, was there any clinical evidence for that occurrence (or references that would indicate that)? Was there any evidence that any of the recombinants were circulating in the community?
Response: Thanks for your advice. In the manuscript, we focused on the identification of recombination. With the method development of distinguishing co-infections from recombinants, we are accumulating clinical data about co-infections or recombinants of HAdVs during 2017 to 2024. We will give another report in the near future.
5.The text in the Figures was too small to be legible (Figures 1, 2, and 5). Even with magnification, they became pixelated and were difficult to read. Perhaps in the published article the print will be more readable, but that may require a better initial figure before the decrease in text size.
Response: We will submit figures in higher pixel.
6.Table 1 needs to be on a single page (in the current manuscript the column headings are on a separate page). It would be helpful to have a better or more extensive description of the categories that are used for Penton. While there is a reference for this, it was in a less accessible journal, so I was not able to see what the differences are.
Response: In the final version, Table 1 will be put on a same page.
For more extensive description of the categories that are used for Penton, we added in 2.3 that “Homologous recombination was clearly involved in the molecular evolution of species A, B, and D, but little is known about the molecular evolution of species C. A genetic recombination analysis was carried out by Mao et al on 201 HAdV-C strains from 9 provinces in China between 2000 and 2016 using three capsid proteins. Mao's data revealed that phylogenetic analysis of the penton base sequences of HAdV-C revealed six genetic groups (labelled as Px1-6), which showed that the penton base had more variation than previously thought. However, multiple newly divergent sequences of species C co-circulated across mainland China [14]. Therefore, phylogenetic trees of species C were constructed more carefully by comparing with those newly identified sequences of species C.”
The reference for this was attached in supplementary documents.
7.In line 265, “breakthrough points” may not be the best term. Are these “break points” or “recombination points” instead? Similarly, in line 279, the term “bone frame” is not clear to me. It this the “backbone” or “framework” or some other description?
Response: We revised “breakthrough points” to “break points” and “bone frame” to “backbone” as you suggested.
8.Although 4 or 5 isolates were Species C recombinants, there is not a suggestion that these had identifiable “changes in biological characteristics”. The more interesting observation was that the Species D recombinant was substantially different from the parental types, and the classification as a new Species D type (HAdV-D115) is a very important part of the paper. It may be getting less attention in the paper than it deserves.
Response: As you suggested, the sentence that “Clinical characters of these strains, especially those unidentified, should be accumulated to make sure if these recombination events brought significant changes in biological characteristics.” was edited to “More evidence should be accumulated to assess clinical effects of these recombination events.”
It’s true that the classification of new species D type (HAdV-D115) is very important for the paper. Therefore, we submitted the genome sequence to GenBank with accession number of OR044915. And we contacted with the Human Adenovirus Working Group, who checked the genome sequence and designed of the novel Adenovirus D human/CHN/S8130/2023/115[P22H8F8] as HAdV D115. In Conclusions of Abstract, we declared that “Especially, there is a novel Adenovirus D human/CHN/S8130/2023/115[P22H8F8] designed as HAdV D115.” We added in Results “This novel HAdV-D genome of CHN-BJ-S1830/2021 identified from the respiratory specimens of a 24-day-old neonate was submitted to GenBank and given an accession number of OR044915. The Human Adenovirus Working Group checked the genome sequence and designated the novel Adenovirus D human/CHN/S8130/2023/115[P22H8F8] as HAdV D115.”
Comments on the Quality of English Language
I have noted a few changes in the text above.
You may want to polish the English usage (with someone whose first language is English). While the text is quite readable, there are some subtle ways in which the language could be improved.
Response: To improve the quality of English language, the final version was polished and edited by professional editors at Editage.

Reviewer 3 Report
Comments and Suggestions for Authors
The article “High-frequency recombination of human adenovirus in hospitalized children with acute respiratory tract infections in Beijing, China” by Fangming Wang and colleagues report on epidemiological and virological characterization agent of pediatric acute respiratory infections between 2015 and 2012 collected from hospitalized children focusing on adenovirus as causative agent. The author describe the frequency of adenovirus infections in ARI patients, the distribution of already known adenovirus types among adenovirus positive cases and describe high frequency of new recombinant strains and identify and molecularly characterize a new adenovirus type, HAdV-D115.
The study is carefully designed, methodically very thorough and well presented in the paper.
I have only the following minor points to consider before publication:
1. I hope that the sequences of the newly sequenced genomes were submitted to the GenBank or other public database. At least, I could find an accession number for the new adenovirus type D115 on the home page of the HAdV Working Group (OR044915), which I have not found in the text. It would be important to include a table (a second supplementary table perhaps) for the GenBank accession numbers of the new sequences, which were generated during this study that the scientific community could selectively use this sequences, if there is special interest.
2. It would be nice to see whether there was, how I suspect, a higher frequency of inconsistently typed strains, if the diagnosis was made by PCR compared to the immunological method, or not. The authors should have the data for this analysis. It is not a main issue, but may give us a hint to the efficiency of the multiplex PCR.
3. It would be great if the authors refer to the supplementary table, which contains the GenBank accession numbers of the reference sequences used in this study in the materials and method section, which describes then sequence analysis (this table is not referred anywhere in the main text either).
4. Lane 158 “For virus cloning purification” should be “For plaque purification,” one can name plaque purified viruses biological clones, but “cloning” is a term more for molecular cloning, therefore for sake of clarity I suggest specifying the method exactly.
5. Please, check the supplementary table. In the file, which I could download, the first two lanes in pages 2-4 are scrambled. I think, it would be a better idea to use a here a pdf format.
Author Response
- I hope that the sequences of the newly sequenced genomes were submitted to the GenBank or other public database. At least, I could find an accession number for the new adenovirus type D115 on the home page of the HAdV Working Group (OR044915), which I have not found in the text. It would be important to include a table (a second supplementary table perhaps) for the GenBank accession numbers of the new sequences, which were generated during this study that the scientific community could selectively use this sequences, if there is special interest.
Response: Thanks for your advice. We added these sentences in the final of Results “This novel HAdV-D genome of CHN-BJ-S1830/2021 was identified from the respir-atory specimens of a 24-day-old neonate, and submitted to GenBank with accession number of OR044915. The Human Adenovirus Working Group checked the genome sequence and designated the novel Adenovirus D human/CHN/S8130/2023/115[P22 H8F8] as HAdV D115.” The genome sequence will be released after the publication of the manuscript.
The raw data for Meta-genomic Next-Generation Sequencing were submitted to GenBank with the number of SRR28980650 (CHN-BJ-1w5060/2019),SRR28980649 (CHN-BJ-93578/2018), SRR28980648 (CHN-BJ-86413/2017), SRR28980647 (CHN-BJ-95031/2018) and SRR28980646 (CHN-BJ-s8130/2021), respectively. Then their genome sequences also were submitted to GenBank which will be released after the publication of the manuscript.
- It would be nice to see whether there was, how I suspect, a higher frequency of inconsistently typed strains, if the diagnosis was made by PCR compared to the immunological method, or not. The authors should have the data for this analysis. It is not a main issue, but may give us a hint to the efficiency of the multiplex PCR.
Response: We used multiplex PCR, combined with plaque purification and Meta-genomic Next-Generation Sequencing to distinguish co-infections from recombinants revised according to method established by McCarthy T. et al. To make the design more clearly, we add figure S1.
Figure S1 An algorithm to distinguish co-infection from recombination using multiplex PCR for penton base, hexon and fiber gene amplification, plaque purification and Meta-genomic Next-Generation Sequencing
- It would be great if the authors refer to the supplementary table, which contains the GenBank accession numbers of the reference sequences used in this study in the materials and method section, which describes then sequence analysis (this table is not referred anywhere in the main text either).
Response: It’s a pity that Table S1 in “2.3. Phylogenetic analysis for HAdV typing”, with the heading of “Table S1 The reference genome sequences downloaded from GenBank” was omitted in submission. We’ll check more carefully to ensure the submission of Table S1.
- Lane 158 “For virus cloning purification” should be “For plaque purification,” one can name plaque purified viruses biological clones, but “cloning” is a term more for molecular cloning, therefore for sake of clarity I suggest specifying the method exactly.
Response: As you suggested, “For virus cloning purification” was edited to “For plaque purification”.

Reviewer 4 Report
Comments and Suggestions for Authors
Wang et al. present a very interesting study over respiratory samples collected over a relatively long period of time (6 years) and covering thousands of samples to identify the adenoviral cases. This study presents interesting approaches to remove any concern about possible coinfections and also potentially could provide the prevalence of specific types in the examined area. However, I recommend that besides an extensive proof-reading edition by a native English speaker, some additional concerns must be addressed in order to provide a highly attractive and useful report to the clinical and scientific community. Following my concerns, which I would appreciate if they can be considered:
1. In the abstract and introduction it is assumed that all recombination events lead to highly pathogenic and infectious novel adenovirus; however, this is not true as much of recombination events pass undetected and in many cases will lead to non-viable viruses. I suggest to revise this statement to the potential risk that a recombination represents to become highly pathogenic or infectious.
2. In the line 53-66 of the introduction, it is stated that 53 to 114 originated from recombination events, and while there is some truth to that, types between 1 and 52 have experienced also recombination events in their origins. So, this paragraph should state better that recombination is a prevalent characteristic and driver of diversity in the genus Mastadenovirus.
3. Line 67 in the introduction states that penton base, hexon and fiber have been reported as recombination hotspots. However, it is worth noting that those are not the only recombination hotspots for the Mastadenovirus genus, as the E3 transcriptional region is also considered a recombination hotspot.
4. The section 2.2 of materials and methods could be easily summarized in a table. Actually at the bottom of that page I see a table S2 (why S2?) where these information of primers and lengths could be included, simplifying the reading and enhancing the comparison of primers.
5. Mafft is a multiple sequence alignment tool, it doesn't measure the divergence as stated in lines 120-121.
6. Lines 121-124 have multiple issues: how the accurate typing is achieved by the phylogenetic trees? What is the criteria? Is clustering with a specific type enough to be considered part of that type? Is there consideration of identity, evolutionary distance or similarity? What parameters are used for the maximum likelihood? What evolutionary method is used in the maximum likelihood method? What is the treatment for gaps and missing data?
7. Lines 125-127 suggests that phylogenetic trees were more carefully constructed for HAdV-C, how? what is the special treatment for these sequences? Why were not the trees for other species also inferred carefully?
8. In section 2.5, although the CPE is clinically the traditional method for isolation of the virus, have any other results such as qPCR being considered to measure the presence of the virus? If so, could the Ct values be added as information of what limits of detection correlate with the isolation possibilities?
9. Section 2.7, the FASTQ files should be made available in a public repository such as SRA. That will make possible to replicate the results or confirm the absence of coinfections by any reader interested on doing more with the data. If the virus are recombinant, what reference sequences were used for the assembly? Please provide accession numbers and how you made sure that didn't affect the resulting consensus sequence.
10. Line 181, technically the sequences analyzed for the recombination evidence were the consensus sequences and not the mNGS, as the mNGS are the reads.
11. Please provide details on what the criteria was to accept the recombination events evidence in section 2.8.
12. Line 191, consider correcting the line from "the ratio of males to females’ 1.36:1" to 'the ratio of males to females was 1.36'
13. In section 3.1, is the information per year available? Consider adding a figure with a barplot chart per year providing the information of the distribution of cases per year among the types. Such information can be very useful to create the picture of the recurrence of these viruses and how they relate from year to year. Maybe the total of cases can be also included in the figure as a pie chart with the percentages to visualize better the distribution of such cases.
14. In section 3.2, only the cases of coinfections were shown in Table 2 (please revise the order of your tables); consider adding a couple of cases tested with negative results for coinfections, in that way how this technique informs your decisions will be clear for the readers and probably replicated in the future. Also, in table 2, add as footnote what is PC
15. Why in section 3.3 hexon is mentioned before penton base? It is convention to describe penton base, hexon and fiber. Also, the scales in figure 1 are different without description in the legend (please improve the legends), this can be considered suspicious.
16. Maybe it is due to PDF conversion, but the quality of the figures is really low and avoids reading into them. Consider increasing the DPI at least for the peer-review process or providing PDFs for your figures.
17. Lines 223-226, state the identity/evolutionary distance of the penton base, hexon and fiber genes to the closest matches for the novel type. Otherwise it is suspicious.
18. As stated for figure 1, in figure 2 the different scales can be suspicious as it seems enhancing distances. Also, it is not stated if the sequences are nucleotides or amino acids. How much is the identity at amino acid level?
19. Table 1 (you see what I meant? The table 1 comes after table S2 and table 2, what is this order?) could be informative if the evolutionary distance or at least identity at nucleotide and amino acid level were provided. Also, at least add a footnote to look into the figures to understand what is Px*
20. Figure 3 lacks information about the number of sequences collapsed in the omitted groups.
21. As neither identity values or the RDP evidence supporting the recombination events are provided, it seems clear that the sequences of the HAdV-C are not recombinants because they clustered just in the middle of Type 2 and Type 5; potential recombination evidence could be false positive results.
22. Line 255, maybe you meant to say breakpoints (breakthrough has a different meaning).
23. It seems that all the recombination events results were based on Simplot (with a huge window size and huge step size), as no evidence from RDP is provided or referenced. Simplot with such huge window size and step size won't produce accurate description of the recombination breakpoints.
24. The analysis intended in Figure 4 should be also evaluated with other sequences to provide context of the null hypothesis. As stated by the authors, lots of recombination events happened in the evolution of Human mastadenovirus types, how can we be sure if the detected recombination events are actually novel recombination events for these sequences or detection of ancient recombination events common to all sequences in that type. This criticism comes from the fact that the four sequences seem to be clearly located in the middle of those types.
25. Type 64 according to the HAWG (http://hadvwg.gmu.edu/) is P22H19F37, so maybe it is more appropriate to name your type 115 as P22H8F8, also following the denomination provided by the HAWG. Again, more can be discussed if you provide the identity values compared against those types.
26. lines 342-351 seems to suggest that you don't agree with keeping the C putative recombinants as classified under C2 and C5, but no evidence is provided for supporting that disagreement. The RDP evidence should help here to provided statistical support to the claims and whether it is a false positive event, or recombination inside the same type, which although interesting could be addressed in a different way.
Comments on the Quality of English Language
Dear editor,
Although the text is readable there are some omissions and grammar issues. Also, the authors were not careful in the quality of the figures and the order of the tables; that bothers me because I feel like I am the first person reading the whole manuscript and noticing those issues.
Apologies if my assessment is harsh.
Best regards,
--
GG
Author Response
- In the abstract and introduction it is assumed that all recombination events lead to highly pathogenic and infectious novel adenovirus; however, this is not true as much of recombination events pass undetected and in many cases will lead to non-viable viruses. I suggest to revise this statement to the potential risk that a recombination represents to become highly pathogenic or infectious.
Response: As you suggested, we revised the sentence “Recombination events often occur to form highly pathogenic and infectious novel human adenovirus (HAdV) types.” to “Therefore, recombination events were common in HAdV, and the main driver of diversity in the genus Mastadenovirus, which have led to some new highly pathogenic or infectious types [6]. It’s vital to monitor recombinant HAdVs, especially in children with acute respiratory tract infections (ARIs).”
- In the line 53-66 of the introduction, it is stated that 53 to 114 originated from recombination events, and while there is some truth to that, types between 1 and 52 have experienced also recombination events in their origins. So, this paragraph should state better that recombination is a prevalent characteristic and driver of diversity in the genus Mastadenovirus.
Response: We revised these sentences to “Among these 114 types, HAdV 52 to HAdV114, which caused inconsistent serological results and misleading typing in classical HAdVs typing by serum neutralization and hemagglutination inhibition tests to detect the antigenic determinants of hexon and fiber, were typed relying on genome sequences. ”, and “Therefore, recombination events were common in HAdV, and the main driver of diversity in the genus Mastadenovirus, which have led to some new highly pathogenic or infectious types [6]. It’s vital to monitor recombinant HAdVs, especially in children with acute respiratory tract infections (ARIs).”
- Line 67 in the introduction states that penton base, hexon and fiber have been reported as recombination hotspots. However, it is worth noting that those are not the only recombination hotspots for the Mastadenovirus genus, as the E3 transcriptional region is also considered a recombination hotspot.
Response: Yes, the E3, E1 and E4 transcriptional region are also considered recombination hotspots. The sentence was edited to “Moreover, penton base, hexon, and fiber genes, as well as the E3, E1 and E4 transcriptional regions, have been reported as the main recombination hotspots [8 ].”
- The section 2.2 of materials and methods could be easily summarized in a table. Actually at the bottom of that page I see a table S2 (why S2?) where these information of primers and lengths could be included, simplifying the reading and enhancing the comparison of primers.
Response: As you suggested, we edited Table S2 to Table 1, and these information of primers and lengths were added in the revised version.
- Mafft is a multiple sequence alignment tool, it doesn't measure the divergence as stated in lines 120-121.
Response: As you advised, we change “The divergence of these sequences were shown in alignment using the MAFFT software. ” to “Multiple sequence alignment was carried out using MAFFT software.”
- Lines 121-124 have multiple issues: how the accurate typing is achieved by the phylogenetic trees? What is the criteria? Is clustering with a specific type enough to be considered part of that type? Is there consideration of identity, evolutionary distance or similarity? What parameters are used for the maximum likelihood? What evolutionary method is used in the maximum likelihood method? What is the treatment for gaps and missing data?
Response: As we described in the revised version that the accurate typing is achieved according to the criteria “The typing of these sequences were determined by constructing phylogenetic trees with penton base, hexon and fiber genes of reference sequences (Table S1) retrieved from GenBank, respectively, according to the maximum likelihood method and 1000 bootstrap pseudo-replicates implemented in MEGA X with pairwise deletion of gaps and missing data. In order to get concise phylogenetic trees, the annual representative sequences of different types were selected from sequences with high identity (>85%) and in one cluster. Homologous recombination was clearly involved in the molecular evolution of species A, B, and D, but little is known about the molecular evolution of species C. A genetic recombination analysis was carried out by Mao et al on 201 HAdV-C strains from 9 provinces in China between 2000 and 2016 using three capsid proteins. Mao's data revealed that phylogenetic analysis of the penton base sequences of HAdV-C revealed six genetic groups (labelled as Px1-6), which showed that the penton base had more variation than previously thought. However, multiple newly divergent sequences of species C co-circulated across mainland China [14]. Therefore, phylogenetic trees of species C were constructed more carefully by comparing with those newly identified sequences of species C.”
The phylogenetic trees were constructed using sequences of penton base, hexon and fiber genes with pairwise deletion of gaps and missing data.
- Lines 125-127 suggests that phylogenetic trees were more carefully constructed for HAdV-C, how? what is the special treatment for these sequences? Why were not the trees for other species also inferred carefully?
Response: To answer the question, we described “Homologous recombination was clearly involved in the molecular evolution of species A, B, and D, but little is known about the molecular evolution of species C. A genetic recombination analysis was carried out by Mao et al on 201 HAdV-C strains from 9 provinces in China between 2000 and 2016 using three capsid proteins. Mao's data revealed that phylogenetic analysis of the penton base sequences of HAdV-C revealed six genetic groups (labelled as Px1-6), which showed that the penton base had more variation than previously thought. However, multiple newly divergent sequences of species C co-circulated across mainland China [14]. Therefore, phylogenetic trees of species C were constructed more carefully by comparing with those newly identified sequences of species C.”
- In section 2.5, although the CPE is clinically the traditional method for isolation of the virus, have any other results such as qPCR being considered to measure the presence of the virus? If so, could the Ct values be added as information of what limits of detection correlate with the isolation possibilities?
Response: In the research work, virus isolation was used to harvest virus strains for plaque assays, rather than to measure the presence of the virus. To make it clear, we edited the sentence to “To harvest HAdV strains for plaque assays, specimens suspected of containing recombinant HAdVs were then inoculated into A549 cells, which were maintained in Dulbecco’s modified Eagle medium (DMEM) supplemented with 2% fetal bovine serum.”
- Section 2.7, the FASTQ files should be made available in a public repository such as SRA. That will make possible to replicate the results or confirm the absence of coinfections by any reader interested on doing more with the data. If the virus are recombinant, what reference sequences were used for the assembly? Please provide accession numbers and how you made sure that didn't affect the resulting consensus sequence.
Response: Thanks for your advice. We will release the FASTQ files of the genome sequences from Meta-Genomic Next-Generation Sequencing after the publication of the manuscript.
“The whole genome sequences of CHN-BJ-86413/2017, CHN-BJ-93578/2018, CHN-BJ-95031/2018, CHN-BJ-1w5060/2019, and CHN-BJ-S8130/2021 determined by mNGS were assembled in non-parametric style, combined with parametric assembly optimization by using MZ151862_RUS- Novosibirsk_8.234V_ 2020, MK041234_Shanxi-CHN_22_2002, MK041237_Shanxi-CHN_38_2007, MK165453_ CHN-SX-2004-327_2004 and FJ169625_ DEU_IAI-1_2005 as reference sequences, respectively, used for phylogenetic analysis. The raw data for Meta-genomic Next-Generation Sequencing were submitted to GenBank with the number of SRR28980650 (CHN-BJ-1w5060/2019),SRR28980649 (CHN-BJ-93578/2018), SRR28980648 (CHN-BJ-86413/2017), SRR28980647 (CHN-BJ-95031/2018) and SRR28980646 (CHN-BJ-s8130/2021), respectively. Then their genome sequences also were submitted to GenBank which will be released after the publication of the manuscript.
- Line 181, technically the sequences analyzed for the recombination evidence were the consensus sequences and not the mNGS, as the mNGS are the reads.
Response: To make it clear, we edited the sentence to “Phylogenetic network analysis was performed using SplitsTree (version 4.14.6) to visualize recombination events based on genome sequences assembled from mNGS.
- Please provide details on what the criteria was to accept the recombination events evidence in section 2.8.
Response: The sentences, “Default settings were used and the threshold p-value set at 0.05 using the Bonferroni correction for the RDP4. To avoid false positive results, recombination events supported by at least six different methods were considered.” were added.
- Line 191, consider correcting the line from "the ratio of males to females’ 1.36:1" to 'the ratio of males to females was 1.36'
Response: Thanks! We corrected it by editing the sentence to “During 2015 to 2021, a total of 16,097 pediatric patients with ARIs were included, in which the ratio of males to females was 1.36, and the median age was 3.00 (1.00-5.95) years.”
- In section 3.1, is the information per year available? Consider adding a figure with a barplot chart per year providing the information of the distribution of cases per year among the types. Such information can be very useful to create the picture of the recurrence of these viruses and how they relate from year to year. Maybe the total of cases can be also included in the figure as a pie chart with the percentages to visualize better the distribution of such cases.
Response: The information per year with a bar plot chart was present in Figure 1, as well as a pie chart with the percentages of HAdV types.
Figure 1. The distribution of HAdV-positive cases in pediatric patients with acute respiratory illness from Jan 2015 to Dec 2021, in Beijing. (A) The yearly distribution of HAdV typing results. (B) Proportions of different HAdV types.
B-others: HAdV-B14, -B21, et al. C-others: HAdV-C2, -C5, C6, et al. Undetermined: specimens with different typing results of the three capsid genes. Unknown: specimens lacking at least one gene sequence among the three capsid genes.
- In section 3.2, only the cases of coinfections were shown in Table 2 (please revise the order of your tables); consider adding a couple of cases tested with negative results for coinfections, in that way how this technique informs your decisions will be clear for the readers and probably replicated in the future. Also, in table 2, add as footnote what is PC
Response: As you suggested, we changed the order of tables. It’s true that only the nine cases with over one kind of fiber gene sequences belonging to different types were shown in Table 2, whilst the other 26 cases with only one fiber gene sequence and suspected of being infected by recombinant were shown in Table 3. The footnote of PC was added as “The type of penton base gene is HAdV-C”.
- Why in section 3.3 hexon is mentioned before penton base? It is convention to describe penton base, hexon and fiber. Also, the scales in figure 1 are different without description in the legend (please improve the legends), this can be considered suspicious.
Response: As you suggested, we changed the order of the three capsid proteins to “penton base, hexon and fiber” in the revised version with the description “(A) the phylogenetic tree of the nucleotide sequences of penton base gene, divided into clusters of Px1Ps1, Px1Ps6, Px1Ps3, Px2, Px3, Px5 and P89. Ps, the subgroup. Px, the major genetic group. (B) the phylogenetic tree of the nucleotide sequences of hexon gene, divided into H1, H2, H57, H6 and H5. (C) the phylogenetic tree of the nucleotide sequences of fiber gene, divided into F2, F5, F1 and F6.”
- Maybe it is due to PDF conversion, but the quality of the figures is really low and avoids reading into them. Consider increasing the DPI at least for the peer-review process or providing PDFs for your figures.
Response: We will try our best to increase the DPI of the figures.
- Lines 223-226, state the identity/evolutionary distance of the penton base, hexon and fiber genes to the closest matches for the novel type. Otherwise it is suspicious.
Response: For 25 cases belonging to species C were excluded from co-infections, it’s difficult to state the identity/evolutionary distance of the penton base, hexon and fiber genes to the closest matches for the novel type. Therefore, we just showed all sequence analysis results in phylogenetic trees, while partial of them were shown in Table S2.
Table S2 The closest match genes and their nucleotide identity (%)
|
Genome seqences |
the closest match penton base gene (identity %) |
the closest match hexon gene (identity %) |
the closest match fiber gene (identity %) |
|
CHN-BJ-86413/2017 |
LC504573(99.76%) |
LC504573(99.94%) |
MN628615(100%) |
|
CHN-BJ93578/2018 |
LC504573(99.59%) |
KF268199(99.81%) |
KF268199(100%) |
|
CHN-BJ-95031/2018 |
KF268129(99.20%) |
KF268127(99.81%) |
KF429754(99.54%) |
|
CHN-BJ-1w5060/2019 |
MZ151865(100%) |
MK883607(100%) |
MK883607(100%) |
|
CHN-BJ-S8130/ |
AB448767(99.75%) |
EF121005(99.94%) |
AB448767(100%) |
- As stated for figure 1, in figure 2 the different scales can be suspicious as it seems enhancing distances. Also, it is not stated if the sequences are nucleotides or amino acids. How much is the identity at amino acid level?
Response: Thanks for your comments. The scales of evolutionary trees were on the basis of the nucleotide sequences which were different on different evolutionary trees. The footprints were labeled in figure 1 and figure 2.
- Table 1 (you see what I meant? The table 1 comes after table S2 and table 2, what is this order?) could be informative if the evolutionary distance or at least identity at nucleotide and amino acid level were provided. Also, at least add a footnote to look into the figures to understand what is Px*
Response: Sorry for our carelessness. We have carefully edited our manuscript and corrected the error in the revised version.
As you suggested, we added in Figure 2 “(A) the phylogenetic tree of the nucleotide sequences of penton base gene, divided into clusters of Px1Ps1, Px1Ps6, Px1Ps3, Px2, Px3, Px5 and P89. Ps, the subgroup. Px, the major genetic group. (B) the phylogenetic tree of the nucleotide sequences of hexon gene, divided into H1, H2, H57, H6 and H5. (C) the phylogenetic tree of the nucleotide sequences of fiber gene, divided into F2, F5, F1 and F6.”
- Figure 3 lacks information about the number of sequences collapsed in the omitted groups.
Response: As you suggested, the number of sequences collapsed in the omitted groups were added in figure 4.
- As neither identity values or the RDP evidence supporting the recombination events are provided, it seems clear that the sequences of the HAdV-C are not recombinants because they clustered just in the middle of Type 2 and Type 5; potential recombination evidence could be false positive results.
Response: Thanks for your advice. We add the date about a summary of possible recombination events identifed by RDP4.To identify the recombination events, recombination analysis was per formed using the RDP4 package with multiple algorithms. Seven algorithms (RDP, GENECONV, BootScan,MaxChi, Chimaera, SiScan, 3Seq) were utilized to predict potential recombination events between the input sequences. Six recombination events were identified with a high level of confidence as recombinants, according to all seven methods implemented in the RDP package. (Table S3)
Table S3. Possible recombination events of four HAdV-C and one HAdV-D strains identifed by RDP4.
|
Recombinant Strain |
Parent Major/Minor |
Recombinant Region in Alignment |
Model (Average P Value) |
||||||
|
RDP |
GENECONV |
Bootscan |
Maxchi |
Chimaera |
SiSscan |
3Seq |
|||
|
CHN-BJ-86413/2017 |
AC_000008/MH558113 |
15010-28171 |
2.686×10^2 |
1.102×10^2 |
3.592×10^3 |
6.455×10^4 |
5.182×10^4 |
7.587×10^12 |
9.690×10^5 |
|
CHN-BJ-95031/2018 |
AF534906/AC_000008 |
27794-31362 |
9.320×10^17 |
1.450×10^53 |
3.186×10^75 |
1.269×10^38 |
1.118×10^25 |
4.251×10^89 |
4.389×10^11 |
|
CHN-BJ93578/2018 |
AC_000008/HQ003817 |
12036-14847 |
4.584×10^20 |
2.483×10^16 |
3.042×10^13 |
9.284×10^16 |
9.540×10^5 |
/ |
3.854×10^13 |
|
CHN-BJ-1W5060/2019 |
NC_001405/AF534906 |
11814-19207 |
5.214×10^53 |
1.317×10^51 |
7.691×10^22 |
2.690×10^20 |
3.340×10^19 |
3.403×10^18 |
1.865×10^14 |
|
CHN-BJ-S8130/2021 |
FJ169625 /LC314153 |
11,444-17,721 |
2.708×10^121 |
1.520×10^127 |
4.229×10^124 |
5.996×10^35 |
1.558×10^35 |
2.197×10^39 |
2.220×10^15 |
|
  |
17,722-20,998 |
3.401×10^187 |
9.744×10^187 |
2.256×10^190 |
6.439×10^39 |
9.486×10^40 |
8.145×10^42 |
6.661×10^15 |
|
- Line 255, maybe you meant to say breakpoints (breakthrough has a different meaning).
Response: We corrected the error (breakpoints) in the revised version.
- It seems that all the recombination events results were based on Simplot (with a huge window size and huge step size), as no evidence from RDP is provided or referenced. Simplot with such huge window size and step size won't produce accurate description of the recombination breakpoints.
Response: Thanks for your advice. The length of the HAdV genome is 34-36 kb. Therefore, the length of DNA sequence with this window size (1000bp) and step size (200bp) were better than the window size (200bp) and step size(20bp).
- The analysis intended in Figure 4 should be also evaluated with other sequences to provide context of the null hypothesis. As stated by the authors, lots of recombination events happened in the evolution of Human mastadenovirus types, how can we be sure if the detected recombination events are actually novel recombination events for these sequences or detection of ancient recombination events common to all sequences in that type. This criticism comes from the fact that the four sequences seem to be clearly located in the middle of those types.
Response: Thanks for your advice. We have checked the results by adding other sequences as control in the recombination analysis as shown in figures below. Compared to the control sequence, the recombination event indeed exists instead of detecting the ancient recombination events common to all sequences in that type.
- Type 64 according to the HAWG (http://hadvwg.gmu.edu/) is P22H19F37, so maybe it is more appropriate to name your type 115 as P22H8F8, also following the denomination provided by the HAWG. Again, more can be discussed if you provide the identity values compared against those types.
Response: We accept your suggestion to name the novel type as HAdV-D115 (P22H8F8). And we revised these sentences to “Therefore, strain CHN-BJ-S8130/2021 was a novel recombinant HAdV with the major region of 11,444-17,721 bp containing the penton base gene originated from HAdV-D64 (P22H19F37), while the minor region of 17,722-20,998 bp containing the hexon gene was derived from HAdV-D8, and the backbone was from HAdV-D53 (P37H22F8) with the fiber gene from AdV-D8.”
- lines 342-351 seems to suggest that you don't agree with keeping the C putative recombinants as classified under C2 and C5, but no evidence is provided for supporting that disagreement. The RDP evidence should help here to provided statistical support to the claims and whether it is a false positive event, or recombination inside the same type, which although interesting could be addressed in a different way.
Response: Thank you very much for your suggestion. We consider the four HAdV-C strains were the intermediate adenovirus strains by the RDP evidence and phylogenetic tree results.

Round 2
Reviewer 4 Report
Comments and Suggestions for Authors
Thank you to the authors for considering my suggestions and addressing my concerns.